Effects of cowpea mild mottle virus on soybean cultivars in Brazil

Barreto da Silva Felipe 1 felipe.barreto@unesp.br
Muller Cristiane 2
http://orcid.org/0000-0002-2869-8119 Bello Vinicius Henrique 1
Watanabe Luís Fernando Maranho 1
http://orcid.org/0000-0002-4401-2594 Rossitto De Marchi Bruno 1
Fusco Lucas Machado 1
Ribeiro-Junior Marcos Roberto 1
Minozzi Guilherme Barbosa 2
Vivan Lucia Madalena 3
Tamai Marco Antonio 4
Farias Juliano Ricardo 5
Nogueira Angélica Maria 1
Sartori Maria Márcia Pereira 1
Krause-Sakate Renate 1 renate.krause@unesp.br
1 Department of Plant Protection, Universidade Estadual Paulista “Julio de Mesquita Filho” (UNESP) , Botucatu, São Paulo , Brazil
2 Corteva™ Agrisciences , Mogi Mirim, São Paulo , Brazil
3 Fundação de Apoio a Pesquisa à Pesquisa Agropecuária de Mato Grosso/Fundação MT , Rondonópolis, Mato Grosso , Brazil
4 Department of Human Sciences, Universidade do Estado da Bahia/UNEB, Campus IX , Barreiras, Bahia , Brazil
5 Department of Entomology, Instituto Phytus , Santa Maria, Rio Grande do Sul , Brazil
Dinesh-Kumar Savithramma
Electronic publication date: 2020 Aug 31
Publication date: 2020
Volume: 8
Electronic Location ID: e9828
Received 2020 Apr 6; Accepted 2020 Aug 5
Copyright: © 2020 Barreto da Silva et al.
Copyright year: 2020
Copyright holder: Barreto da Silva et al.
License: This is an open access article distributed under the terms of the Creative Commons Attribution License, which permits unrestricted use, distribution, reproduction and adaptation in any medium and for any purpose provided that it is properly attributed. For attribution, the original author(s), title, publication source (PeerJ) and either DOI or URL of the article must be cited.
License URL: https://creativecommons.org/licenses/by/4.0/

Keywords: CPMMV, Stem necrosis, Soybean disease, Transmission

Funding: Coordenação de Aperfeiçoamento de Pessoal de Nível Superior, Brasil (CAPES) 001 Fundação de Amparo à Pesquisa do Estado de São Paulo (FAPESP) 2017/21588-7 Corteva™ Agrisciences CNPq scholarships CNPq research fellow This project was financially supported by the Coordenação de Aperfeiçoamento de Pessoal de Nível Superior, Brasil (CAPES)—Finance Code 001 and Fundação de Amparo à Pesquisa do Estado de São Paulo (FAPESP—process number 2017/21588-7) and by grants from Corteva™ Agrisciences. Felipe Barreto da Silva is recipient of CNPq scholarships. Renate Krause-Sakate is a CNPq research fellow. The funders had no role in study design, data collection and analysis, decision to publish, or preparation of the manuscript.

==============================
Soybean stem necrosis is caused by cowpea mild mottle virus (CPMMV), transmitted by the whitefly Bemisia tabaci. CPMMV has already been recorded in all major soybean-producing areas of Brazil. The impacts caused by CPMMV to the current Brazilian soybean production are unknown, thus the main objective of this study was to evaluate the effects of CPMMV infection on the main important soybean cultivars grown in the Southern and Midwestern regions of Brazil. Although asymptomatic in some of the tested cultivars, CPMMV infection significantly reduced the plant height, the number of pods per plant and the 1,000-grain weight. In addition, estimated yield losses ranged from 174 to 638 kg ha−1, depending on the cultivar. Evidence of seed transmission of CPMMV was observed in the BMX POTÊNCIA RR cultivar. These results suggest that CPMMV could have an important role in the reduction of soybean productivity in Brazil, but symptomless infections might be hiding the actual impact of this pathogen in commercial fields and infected seeds could be the primary inoculum source of the virus in the field.

Introduction

Soybean (Glycine max (L.) Merril) is an important crop worldwide as a source of oilseed and protein. Brazil is the second largest producer of soybean in the world, producing 114.8 million tons in a cultivated area of 35.8 million hectares in the 2018/2019 growing season. In Brazil, soybean is the most important economic crop, generating approximately 675 million US$ to the internal market and 31 billion US$ to exportation (Hirakuri & Lazzarotto, 2014; Conab, 2018). In the last decade, the soybean cultivated area in Brazil increased 64.9%, while the productivity increased from 2,800 to 3,400 kg ha−1 over the same time (Conab, 2018).

Soybean can be affected by several pests regardless of growth stage. The occurrence of at least 15 viral diseases have been reported in Brazil (Almeida, 2008; De Marchi et al., 2018). The Cowpea mild mottle virus (CPMMV; family Betaflexiviridae, genus Carlavirus), which is the agent of soybean “stem necrosis disease”, is a single-stranded positive sense RNA virus with flexuous filamentous particles (approx. 650 nm in length). The genome of 8,200 nucleotides has a cap structure (m7GpppG) linked to the 5′ terminus and a polyadenylated tail at the 3′ end (Menzel, Winter & Vetten, 2010; King et al., 2011; Zanardo et al., 2014a) and has a genomic organization with six open reading frames (ORFs), typical of the genus Carlavirus. ORF1 encode a replicase protein containing four conserved motifs: methyltransferase, C23 peptidase, RNA helicase and an RNA-dependent RNA polymerase. ORF2, ORF3 and ORF4 encode proteins of the triple gene block (TGB). ORF5 encodes the coat protein (CP), and ORF6 encodes a nucleic acid binding protein (Menzel, Winter & Vetten, 2010). CPMMV was reported infecting soybean in the 2000/2001 season in the state of Goiás and having been subsequently identified in soybean fields across Brazil in the states of Bahia, Mato Grosso, Maranhão, Paraná (Almeida et al., 2003, 2005; Almeida, 2008) and in 2008, in the Minas Gerais and Tocantins states (Almeida, 2008). Although steam necrosis is the common name of this disease, in the last years, mild mottle and mosaic were the most common symptoms observed for this virus infection (Zanardo et al., 2014a; Zanardo & Carvalho, 2017). The use of resistant cultivars is the most important method to reduce losses caused by virus disease. Resistant soybean cultivars to CPMMV have been reported in India (cv. F4C7-32 and JS335) (Cheruku et al., 2017), in Puerto Rico (cv. IA3023) (Brace, Fehr & Graham, 2012), Indonesia (cv. MLG0120) (Suryanto et al., 2014) and in Brazil (cv. BRS 133) a former and obsolete cultivar (Arias et al., 2015; De Oliveira et al., 2018). In general, carlaviruses are transmitted by aphids (King et al., 2011), however CPMMV is one of the two exceptions of the genus that are transmitted in a non-persistent manner by the whitefly Bemisia tabaci (Gennadius) (Hemiptera: Aleyrodidae) (Almeida, 2008; Marubayashi, Yuki & Wutke, 2010). This pest itself can reduce soybean productivity (Lourenção, Yuki & Alves, 1999; Tamai, Martins & Lopes, 2006) and was listed between the most important pest affecting this crop in Brazil (Brasil, 2019). Bemisia tabaci is also an excellent vectors of viruses (Navas-Castillo, Fiallo-Olivé & Sánchez-Campos, 2011; Gilbertson et al., 2015), affecting several crops such as vegetables, fibers and ornamentals (De Barro et al., 2011; Lapidot et al., 2014). Bemisia tabaci is widely spread in Brazil, and the Middle East Asia Minor 1 (MEAM1, known as B biotype) is the prevalent species on major crops across the country (Moraes et al., 2018). The species Mediterranean (MED, also known as biotype Q), which was firstly reported in the South region of our country (Fonseca Barbosa et al., 2015) was also detected in different states of Brazil associated with ornamental plants, and more rencently, MED was found in greenhouses and open field in the Southeastern of Brazil (Moraes et al., 2017, 2018; Bello et al., 2020).

Cowpea mild mottle virus is also easily transmitted by sap (Brunt & Kenten, 1973), one characteristic that is very helpful to study the virus. The seed transmission of CPMMV has also been reported in different plant species such as soybean, cowpea (Vigna unguiculata) and common bean (Phaseolus vulgaris) in Africa (Brunt & Kenten, 1973), yardlong bean (Vigna unguiculata subsp. sesquipedalis) in Venezuela (Brito et al., 2012), and by some soybean cultivars in India (Yadav et al., 2013). There is no information about soybean seed transmission of Brazilian CPMMV isolates (Almeida et al., 2005).

The effects caused by CPMMV to Brazilian soybean production have never been estimated (Zanardo & Carvalho, 2017), and the ability of Brazilian CPMMV to be transmitted by seed is still unknown. Thus, the goal of this study was to evaluate the damage of CPMMV on the major soybean varieties used in important growing areas of the Southern and Midwestern regions of Brazil. Additionally, the seed transmission ability of one CPMMV isolate from the São Paulo State was assessed.

Materials and Methods

Obtaining CPMMV isolate, complete genome characterization and Bayesian phylogenetic analysis

The CPMMV isolate was collected from soybeans in Casa Branca County, São Paulo State, Brazil (2016/2017). Total RNA was extracted from the leaf tissue of symptomatic soybean plants using the PureLink Viral RNA/DNA Mini Kit (Thermo Fisher Scientific, Waltham, MA, USA) following the manufacturer instructions. A transcription-polymerase chain reaction (RT-PCR) One Step using AMV reverse transcriptase (Promega, São Paulo, Brazil) was performed using the specifical primers CPMMV 1280-F (5′-GGC GTT CCA AAA GCT GCC GAT-3′) and CPMMV 1696-R (5′-GGA GCC ACC TTT CCA ATC AA-3′) (De Marchi et al., 2017). All amplifications consisted of an initial step of 42 °C for 30 min, a second step of 94 °C for 2 min, 30 cycles of 94 °C for 54 s, annealing at 54 °C for 50 s and elongation at 72 °C for 50 s, followed by a final extension step at 72 °C for 10 min. In order to obtain the complete genome characterization of the CPMMV isolate from Casa Branca—SP (called CPMMV Casa Branca_BR), the RNA was used for construction of a cDNA library using the Complete ScriptSeq Kit (Epicenter; Illumina, San Diego, CA, USA) and transcriptome sequencing with Illumina HiSeq2500 platform (Roossinck, Martin & Roumagnac, 2015) at the Center of Functional Genomics (ESALQ/USP, Piracicaba, Brazil). Adapter sequence removal and quality trimming were performed with CLC Genomics Workbench software version 9.0.3. The sequence obtained was analyzed using the software Geneious v11.1.5 (Kearse et al., 2012) and compared to a dataset composed of eight CPMMV complete genome isolates from Ghana (Menzel, Winter & Vetten, 2010), Florida (Rosario et al., 2014) different Brazilian CPMMV isolates described by Zanardo et al. (2014a) and a sequence from India retrieved from GenBank. The sequences were compared using MAFFT v7.222 within the Geneious v.11.1.5 software, and phylogenetic analysis was performed using MrBayes 3.2.2 (Ronquist & Huelsenbeck, 2003). Two independent runs were conducted simultaneously using 10 million generations and excluding 25% from the resulting tree as burnin. Phylogenetic tree was visualized, edited and rooted using FigTree v1.4.4 (tree.bio.ed.ac.uk/software/figtree/). Pairwise comparison between the sequences were performed with the program SDT v.1.2 (Muhire, Varsani & Martin, 2014) using the MUSCLE alignment option (Edgar, 2004). The CPMMV CP (coat protein) nt sequence obtained in this study were also compared with 33 sequences of CPMMV CP retrieved from GenBank, the phylogenetic analysis of the CP can be found in Fig. S1. After virus identification and characterization, the isolate was maintained in common bean (Phaseolus vulgaris L.) cv. Jalo by whitefly transmission. Virus transmission was performed by transferring whitefly specimens (MEAM1) in cages containing infected soybean leaves for a viral acquisition access period (AAP) of 24 h. Following virus acquisition, whiteflies were transferred to cages containing healthy bean plants at the VC (cotyledon leaves) growth stage, for a 24-h inoculation access period (IAP) under controlled conditions at 30 °C. After inoculation, insecticides (Oberon and Cartap) were sprayed on plants to eliminate all the whitefly adults, nymphs and eggs. Thirty days after the IAP, plants were analyzed for the virus presence. After virus confirmation, the plants were used as source of inoculum for the field inoculation experiments.

Experimental areas

Field experiments were conducted during the 2017/2018 growing season with six cultivars distributed in four different growing areas: cv. BMX POTÊNCIA RR in Botucatu, State of São Paulo (coordinates: 22°48′25.4″S, 48°25′46.4″W, elevation: 739 m, sowing date 01/11/2017), cv. M 6410 IPRO and TMG 7062 IPRO in Mogi Mirim, State of São Paulo (coordinates: 22°26′42.8″S 47°04′10.8″W, elevation: 687 m, sowing date 15/12/2017), cv. M 7739 IPRO and M 8372 IPRO in Pedra Preta, State of Mato Grosso (coordinates: 16°50′30.3″S 54°02′39.8″W, elevation: 744 m, sowing date 23/11/2017), and cv. M 9144 RR in Planaltina, Federal District (coordinates: 15°39′51.7″S 47°20′02.0″W, elevation: 887 m, sowing date 27/11/2017). The maximum, minimum and average temperatures and rainfall that occurred during the experimental periods in the four areas were collected from meteorological stations located next to the experimental fields and are available as a climograph in Table S2. There is no technical information about susceptibility/resistance to CPMMV available for all tested cultivar.

The four sites represented some important soybean producing areas in Brazil. They have contrasting environmental conditions, and the cultivars were selected according to the frequency that they were planted in each region. Parameters such as the occurrence of weeds, diseases, and insect pressure, especially B. tabaci, were monitored throughout the season. Field experiments were approved by the Universidade Estadual Paulista Julio de Mesquita Filho (UNESP) and Fundação de Estudos Agrícolas e Florestais (FEPAF) Processo 1259 Dow 01 Renate Krause-Sakate.

Experimental design and field inoculation

The experimental design was a randomized block, with two treatments (healthy and infected plants with CPMMV) with five replications. Each plot was comprised of six rows (5 m), 0.45 m between rows and an average of 14 plants per meter (around 200 soybean plants, totalizing around 1,000 plants/treatment).

Soybean plants were inoculated with CPMMV 30 days after sowing using as inoculum source leaves of common bean cv Jalo infected with CPMMV. Leaves were ground in phosphate buffer 0.01 M, pH 7 containing the abrasive carborundum (600 mesh). The presence of the virus was detected by RT-PCR using specific primers for CPMMV as described previously.

The susceptibility of all cultivar was also tested under the same open field conditions. Approximately 100 seeds per cultivar were sowed in the field in Botucatu during the 2019/2020 growing season, and the seedlings were inoculated 10 days after emergence. Inoculation and virus detection were performed as described previously.

Field sampling and evaluation of agronomic traits

A total of 100 soybean samples collected for each treatment were evaluated for CPMMV infection 30 days after inoculation. When the plants were at physiological maturity (R8), the plant height was evaluated by the distance from the soil to the apex of the plant (cm) and the number of pods per plants was obtained by counting the total number of pods per plant. At the harvest, all plants of the plot were hand harvested and run through a thresher. For each plot, the 1,000-grain weight (g) was determined, which was obtained by weighing 1,000 grains from the plants in the plot, and adjusting to 13% moisture in addition to grain productivity, which was obtained by weighing the grains produced, and adjusting to 13% moisture, then converting into kg ha−1.

CPMMV transmission by soybean seeds

To study the seed-borne capacity of CPMMV, a random sample of seeds were collected from the CPMMV-infected BMX POTENCIA RR plot, harvested in the Botucatu field. These soybean seeds were planted in Styrofoam seedling trays containing Tropstrato HA Hortaliças (Vida Verde Tecnologia em Substratos, Mogi Mirim, SP). The seedling trays were kept in an insect-proof cage. Germination was greater than 90%, and the seedlings did not show any typical disease symptoms. For virus detection, leaf samples were tested using RT-PCR. To compose a sample, leaves of ten plants were collected and combined in 80 samples tested, totaling 800 plants analyzed. Once the presence of CPMMV was detected in a sample, the ten individually plants were tested for the presence of CPMMV. The positive plants were kept in an insect-proof cage for 60 days in order to observe the appearance of symptoms.

Data analysis

Because of the interdependency and interrelationship of agronomic traits, principal component analysis (PCA) was performed to investigate the data collected in the current study. PCA was performed using Minitab 17 Statistical Software (2010). Data were also submitted to analysis of variance (ANOVA) using Statview software (Concepts & StatView, 1987) to determine whether significant differences (p < 0.05) occurred between treatments. Then means were compared using Tukey’s test (α = 5%).

Results

Virus characterization and phylogenetic analysis

Based on pairwise sequence comparision, the complete genome of the CPMMV isolate Casa Banca_BR GenBank accession number MT473963 obtained in this study showed 99% of nucleotide identity with the KC774020—Bean (FL_USA), KC884245—Soybean (Brazil_MG), KC884244—Soybean (Brazil_MG) and KC884246—Soybean (Brazil_MT). According to the classification used by Zanardo et al. (2014b), the CPMMV Casa Branca_BR isolate belongs to the BR2 group, which encompasses the most common CPMMV strains found in soybean in Brazil. This isolate was used as inoculum source for virus infection in all field experiments.

The CPMMV phylogenetic tree of the complete nucleotide genome sequence analysis grouped the CPMMV Casa Branca_BR Isolate with five isolates from Brazil and one from The USA (Fig. 1). Although there are few CPMMV complete sequences published in GenBank, the analysis showed that the isolate used in this study is representative to the Brazilian isolates.

Figure 1 Pairwise identity analysis and phylogenetic analysis.

Genome pairwise identity based on complete sequence of different CPMMV isolates available in GenBank using SDT v1.2. (B) Phylogenetic tree based on the complete sequence of different CPMMV isolates available in GenBank using Bayesian inference (implemented in MrBayes V.3.1, with model GTR+I+G and 10 million generations). Cucumber vein-clearing virus (CuVCV; genus Carlavirus, family Betaflexiviridae) was used as outgroup.

Virus incidence and symptoms on inoculated plots

In the field assays, where the cultivars were planted according to their regions, the symptoms observed were variable among the cultivars. BMX POTÊNCIA RR, M 7739 IPRO and M 8372 IPRO cultivars were symptomless to CPMMV infection. In contrast, the M 6410 IPRO, TMG 7062 IPRO and M 9144 RR cultivars showed the most severe symptoms. The most common symptoms were chlorosis, mottling and mild symptoms (Fig. 2). Due to the variation of symptoms, the estimate of virus infection was carried out by sampling 100 soybean leaves from the plots followed by molecular analysis. For all the six sap-inoculated soybean cultivars, the incidence of CPMMV was greater than 70% (Table 1). The virus transmission efficiency was 70% in cv. M 8372 IPRO, 85% in cv. BMX POTÊNCIA RR, 87% in cv. M 9144 RR, 89% in cv. M 7739 IPRO, 90% in cv. TMG 7062 IPRO and 92% cv. M 6410 IPRO.

Figure 2 Symptoms observed.

Symptoms observed in the six cultivars tested in the field experiment: (A) BMX POTÊNCIA RR, (D) M 7739 IPRO and (E) M 8372 IPRO showed to be symptomless; (C) M 7062 IPRO, (B) M 6410 IPRO and (F) M 9144 RR showed chlorosis, mild and mottling symptoms.

Table 1 Percentage of infeceted plant and mean of agronomic traits of fields assays according their cultivar.

Location	Cultivar	Infected plants (%)	Plant height (cm)	Pods per plant	1,000-grain weight (g)	Yield (kg ha−1)	Yield loss (kg ha−1)	
Healthy	CPMMV-infected	Healthy	CPMMV-infected	Healthy	CPMMV-infected	Healthy	CPMMV-infected	
Botucatu	BMX POTÊNCIA RR	85	78.80a	60.60b	82.10a	40.30b	149.60a	132.60b	4,029a	3,391b	638	
Mogi Mirim	M 6410 IPRO	92	68.71a	59.71b	70.15a	41.26b	139.37a	129.13b	2,564a	2,330b	234	
TMG 7062 IPRO	90	67.13a	66.86a	34.22a	30.92b	186.05a	169.01b	2,393a	2,145b	248	
Pedra Preta	M 7739 IPRO	89	74.76a	70.84b	45.68a	42.89a	145.54a	144.97a	3,323a	2,953a	370	
M 8372 IPRO	70	86.84a	84.48a	50.82a	44.40b	123.66a	119.00a	3,230a	3,056a	174	
Planaltina	M 9144 RR	87	123.13a	121.07a	61.60a	42.45b	148.71a	105.45b	2,278a	1,962b	490	
Note:

Mean followed by the same letter within rows indicate no significant (p < 0.05) difference between healthy and CPMMV-infected plants according to ANOVA.

Symptoms caused by CPMMV Casa Branca_BR isolate in soybean cultivars were slightly similar under the same field conditions (Fig. 3). Symptoms were observed 30 days post-inoculation. BMX POTÊNCIA RR, M 7739 IPRO and M 8372 IPRO cultivars showed mild mottled symptoms, M 6410 IPRO and TMG 7062 IPRO and the other cultivars showed mottle symptoms and the cultivar M 9144 RR showed weak mosaic.

Figure 3 Same condition assay.

Susceptibility of cultivar under the same open field conditions in Botucatu—SP. (A) BMX POTÊNCIA RR, (D) M 7739 IPRO and (E) M 8372 IPRO showed mild mottle symptoms; (B) M 6410 IPRO and (C) M 7062 IPRO showed mottle symptoms and (F) M 9144 RR showed weak mosaic.

Field plot experiments and agronomic performance of the cultivars

The PCA of soybean cultivars comparing CPMMV-infected and healthy plants showed that the proportion of the variance retained by the first principal component (PC1) was 67.5% and for the second principal component (PC2) corresponded to 22.3% of the original remaining variance (Fig. 4).

Figure 4 Principal component analysis (PCA) of the agronomic traits.

Projection of vectors of traits: plant height in cm, number of pods per plants, 1,000-grain weight in g and productivity in kg ha−1.

The exploratory analysis allowed the evaluation of the virus influence on the groups of cultivars and cultivated areas (Figs. 5 and 6). Among the cultivars, the cv. M 9144 RR healthy plants had the greatest plant height and were among the varieties that have the highest number of pods per plants, which set this cultivar and treatment apart from the rest. Regarding the 1,000-grain weight, the cultivar that showed the best performance was TMG 7062 IPRO followed by M 7739 IPRO, M 6410 IPRO and BMX POTÊNCIA RR. The greatest productivity was reached by the cultivar BMX POTÊNCIA RR, followed by M 7739 IPRO and M 8372 IPRO, being the cv. BMX POTÊNCIA RR the most affected by the presence of the virus.

Figure 5 Principal component analysis (PCA) of the agronomic traits.

Biplot graph with dispersion of six soybean cultivars according to the principal components (PC1 and PC2). ⚫ BMX POTÊNCIA RR healthy plants, ○ BMX POTÊNCIA RR CPMMV-Infected plants, ◼ M 6410 IPRO healthy plants, ◻ M 6410 IPRO CPMMV-Infected plants, ♦ TMG 7062 IPRO healthy plants, ◊ TMG 7062 IPRO CPMMV-Infected plants, ▴ M 7739 IPRO healthy plants, ▵ M 7739 IPRO CPMMV-infected plants, ► M 8372 IPRO healthy plants, > M 8372 IPRO CPMMV-infected plants, ◄ M 9144 RR healthy plants, < M 9144 RR CPMMV-infected plants.

As expected, each cultivar had distinct performance once that they have particular characteristics and they were cultivated in areas with contrasting environmental conditions (Fig. 5). The exploration of the data also demonstrated that the cultivars M 6410 IPRO and TMG 7062 IPRO cultivated in Mogi Mirim—SP had a close performance to all evaluated traits as well as the cultivars M 7739 IPRO and M 8372 IPRO which were cultivated in Pedra Preta—MT. The cv. M 9144 RR tested in Planaltina—DF showed the most distant performance data comparing to the other cultivars (Fig. 6).

Figure 6 Principal component analysis (PCA) of the agronomic traits.

Biplot graph with dispersion of four areas according to the principal components (PC1 and PC2). ⚫ Botucatu, São Paulo, ◼ Mogi Mirim, São Paulo, ♦ Pedra Preta, Mato Grosso, ▴ Planaltina, Federal District.

Analysis of variance showed that cultivar tested in this work presented different response to the virus infection, showing a significant (p < 0.05) or non-significant reduction in the traits evaluated (Table 1).

There was a significant effect (p < 0.05) of CPMMV-infection in all agronomic traits evaluated in the cv. BMX POTÊNCIA RR. CPMMV-infected plants have reduction in plant height (p < 0.01, F = 467.76), number of pods per plant (p < 0.01, F = 36.53) and 1,000-grain weight (p < 0.01, F = 11.64) that reflected directly in the productivity (p = 0.01, F = 11.20), which had a loss of 638 kg ha−1, this cultivar being the one that had the greater yield loss, approximately 16%.

Regarding the cultivar M 6410 IPRO, there was also a significant reduction (p < 0.05) in all traits evaluated, and only the plant height (p = 0.74, F = 0.1) did not differ to the cv. TMG 7062 IPRO. The cultivars tested in Pedra Preta—MT differed significantly only in one parameter each, plant height (p < 0.01, F = 17.92) and number of pods per plant (p < 0.01, F = 8.97), for cv. M 7739 IPRO and M 8372 IPRO, respectively. Although no significant effect of treatment in the yield occurred in both cultivars, there was a reduction of 370 kg ha−1 for cv. M 7739 IPRO (p = 0.07, F = 4.08), and 174 kg ha−1 for cv. M 8372 IPRO (p = 0.33, F = 1.07).

All traits in the cv. M 9144 RR, except for plant height, were affected significantly (p < 0.05) due to CPMMV-infection. Although the height was significantly the same for the treatments (p = 0.51, F = 0.47), the reduction caused in number of pods per plant (p = 0.25, F = 7.57) and 1,000-grain weight (p < 0.01, F = 182328.14) in the diseased plants directly affected the in productivity (p = 0.01, F = 9.85) with a reduction of 316 kg ha−1, or approximately 14%.

CPMMV transmission by soybean seeds

From 800 seedlings obtained from seeds harvested in the cv. BMX POTÊNCIA RR CPMMV-infected field plot, three plants were found to be infected by CPMMV, confirmed by RT-PCR. Not a single plant developed typical disease symptoms after emergence until their senescence. The observed percentage of plants infected with the virus was 0.375%, but it is important to mention that the incidence of the virus in these plots was around 85% and seeds from healthy soybean plants were part of the sample.

Discussion

The data obtained in this study revealed that CPMMV causes reduction of productivity, plant height, 1,000-grain weight and pods per plant in the main soybean cultivars used in Brazil, suggesting that this virus may be responsible for economic losses for soybean crop in our country. Additionally, the CPMMV-seed transmission data for a Brazilian isolate highlights the seed importance as a primary inoculum source in the field, especially in areas with low whitefly population, such as the southern states of the country.

Cowpea mild mottle virus was first recorded in Brazil in common bean in the 1980s (Costa, Gaspar & Vega, 1983), and reported as a threat to soybean production in 2002 (Almeida et al., 2003). The losses reported at that time were higher than 85%, since the cultivars used developed the stem necrosis symptom that affected the whole plant (Almeida et al., 2003). The new cultivars has reduced the impact of this disease in soybean since there is no longer the development of systemic necrosis but infected plants show symptoms of mottling and mosaic (Arias et al., 2015). Although some soybean cultivars are symptomless when infected by CPMMV, our results provide evidence that the productivity can be affected. In our study, the highest reduction in productivity was observed for cv. BMX POTÊNCIA RR (638 kg ha−1), a cultivar that did not show any visual symptoms of CPMMV infection (Table 1; Fig. 3). The soybean genotype M 9144 RR, also asymptomatic for CPMMV infection, showed a reduction in productivity around 316 kg ha−1 (Table 1). It is also important to highlight that even a reduction in productivity of 174 kg ha−1 observed for cv. M 8372 IPRO (asymptomatic for CPMMV infection) may cause an economic impact considering that an infected soybean field can show a reduction of productivity around 3 bags ha−1, the bag (60 kg) price is, on average, 20.00 US$ (CEPEA, 2019). As Brazil is the largest soybean oilseed exporter in the world (USDA, 2019), the amount of bags reduction impacts directly not only for the farmers, but also the Brazilian economy.

The seed-borne virus transmission can also be an important component of the epidemiology of the disease in the field. It has already been reported that different isolates of CPMMV can be seed transmitted, as observed for cowpea, soybean and common bean seeds in Ghana (Brunt & Kenten, 1973) and yardlong bean seeds in Venezuela (Brito et al., 2012). In Thailand, the virus was observed to be transmitted by soybean seeds at a frequency lower than 1% (Iwaki et al., 1982) but in India, the seed-borne nature of the virus was detected in several soybean cultivars with higher rates of transmission, ranging from 0.62% to 14.2% (Yadav et al., 2013). Here we provide evidence that Brazilian CPMMV soybean isolates can be seed transmitted. In the world scale, phylogenetic analysis of the CP amino acids sequences demonstrate that CPMMV isolate from Casa Branca clusters together with CPMMV isolate from Argentina (KP402890) and Florida (KC774020), indicating a common origin (Zanardo & Carvalho, 2017). Our data reinforce that the CPMMV capacity to be transmitted by seeds might have contribute for virus dissemination through different countries, highlighting the importance of studying the transmission capacity of this virus by infected seeds.

In addition, a few infected seeds can provide enough CPMMV inoculum to be disseminated by the efficient vector B. tabaci that is considered one of the main important pest for soybean, common bean, melon and tomatoes in Brazil (Brasil, 2018) and a excelent vector of begomovirus, carlavirus and crinivirus (Navas-Castillo, Fiallo-Olivé & Sánchez-Campos, 2011; Gilbertson et al., 2015). Bemisia tabaci MEAM1 (biotype B) is the predominant species in soybean in our country (Moraes et al., 2018) and is a highly polyphagous insect that can feed on more than 600 species of plants (Polston, De Barro & Boykin, 2014). The Brazilian middle-western region is the largest soybean and common bean producer, and both crops are cultivated near to each other. It is already known that whiteflies can colonize soybean and common bean, as well the CPMMV can infect both crops (Marubayashi, Yuki & Wutke, 2010; Inoue-Nagata et al., 2016). The combination of these conditions may contribute for CPMMV transmission, since the common beans can serve as inoculum source of whiteflies and CPMMV to the soybean crop and vice versa.

Conclusion

In summary, we conclude that even asymptomatic for some important soybean genotypes currently planted in Brazil, the CPMMV infection can affect the soybean yield. Seed transmission of the virus can also be an important component for CPMMV dissemination in the field. Soybean breeding programs need to take into account CPMMV infection in order to provide genotypes that are resistant/tolerant to the virus.

Supplemental Information

Supplemental Information 1 CPMMV CP phylogenetic analysis.

Phylogenetic tree of the nt sequences of the CPMMV coat protein (cp) GenBank using Bayesian inference (implemented in MRBAYES V. 3.1, with model GTR+I+G and 10 million generations). The analysis displayed two defined cluster. The CPMMV Brazil Soybean Casa Branca_BR isolate is included with isolates from Argentina (1), Brazil (20), India (5), Puerto Rico (1) Taiwan (2), USA (2) and Ghana (1). Cucumber vein-clearing virus (CuVCV; genus Carlavirus, family Betaflexiviridae) was used as outgroup.

Click here for additional data file.

Supplemental Information 2 Weather data.

Monthly maximum, minimum and average temperature, rainfall occurring during the field experiments in all tested area (Source: Instituto Nacional de Meteorologia do Brasil - INMET).

Click here for additional data file.

Supplemental Information 3 Raw weather data.

Each tab contains the raw data collected for each field and the data about the season 2017/2018.

Click here for additional data file.

Supplemental Information 4 The raw data show all data collected from soybean plants at the harvest moment for each cultivar.

These data were used for statistical analysis to compare the agronomic traits for each cultivar.

Click here for additional data file.

Additional Information and Declarations

Competing Interests

Author Contributions

Field Study Permissions

DNA Deposition

Data Availability

Cristiane Muller and Guilherme Barbosa Minozzi are employed by Corteva™ Agrisciences.

Felipe Barreto da Silva conceived and designed the experiments, performed the experiments, analyzed the data, prepared figures and/or tables, authored or reviewed drafts of the paper, and approved the final draft.

Cristiane Muller conceived and designed the experiments, performed the experiments, analyzed the data, authored or reviewed drafts of the paper, and approved the final draft.

Vinicius Henrique Bello conceived and designed the experiments, performed the experiments, analyzed the data, authored or reviewed drafts of the paper, and approved the final draft.

Luís Fernando Maranho Watanabe conceived and designed the experiments, performed the experiments, analyzed the data, authored or reviewed drafts of the paper, and approved the final draft.

Bruno Rossitto De Marchi conceived and designed the experiments, performed the experiments, analyzed the data, authored or reviewed drafts of the paper, and approved the final draft.

Lucas Machado Fusco conceived and designed the experiments, performed the experiments, analyzed the data, authored or reviewed drafts of the paper, and approved the final draft.

Marcos Roberto Ribeiro-Junior conceived and designed the experiments, performed the experiments, analyzed the data, authored or reviewed drafts of the paper, and approved the final draft.

Guilherme Barbosa Minozzi analyzed the data, authored or reviewed drafts of the paper, and approved the final draft.

Lucia Madalena Vivan analyzed the data, authored or reviewed drafts of the paper, and approved the final draft.

Marco Antonio Tamai analyzed the data, authored or reviewed drafts of the paper, and approved the final draft.

Juliano Ricardo Farias analyzed the data, authored or reviewed drafts of the paper, and approved the final draft.

Angélica Maria Nogueira analyzed the data, prepared figures and/or tables, authored or reviewed drafts of the paper, perfomed phylogenetical and SDT analysis, and approved the final draft.

Maria Márcia Pereira Sartori conceived and designed the experiments, analyzed the data, authored or reviewed drafts of the paper, performed the statystical analysis, and approved the final draft.

Renate Krause-Sakate conceived and designed the experiments, performed the experiments, analyzed the data, authored or reviewed drafts of the paper, and approved the final draft.

The following information was supplied relating to field study approvals (i.e., approving body and any reference numbers):

Field experiments were approved by the Universidade Estadual Paulista Julio de Mesquita Filho (UNESP) and Fundação de Estudos Agrícolas e Florestais (FEPAF) (Processo 1259 Dow 01 Renate Krause-Sakate).

The following information was supplied regarding the deposition of DNA sequences:

Data is available at GenBank: MT473963.

The following information was supplied regarding data availability:

The raw measurements and phylogenetical analysis data are available in the Supplemental Files.

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
