# Peer review of "Effects of cowpea mild mottle virus on soybean cultivars in Brazil"

_PeerJ, doi:10.7717/peerj.9828_

## Round 0.1 · original submission · Major Revisions

Reviewer 1 has raised some interesting points. These should be carefully considered along with Reviewer 2 comments while revising the manuscript.

Reviewer 1 ·

Basic reporting

Authors addressed the effect of CPMMV isolate from Brazil on few soybean cultivars in different geographical locations. The result showed that CPMMV infection reduced crop yield and affected other agronomical traits, especially when a cultivar doesn’t show clear symptoms of infection.


The introduction need more details in terms of the following:
1) I suggest authors to add more description about the Cowpea mild mottle virus: ssRNA(+), genome length, virion types, ORFs encoded.. etc.
2) Also, adding literature about soybean cultivars that have been reported to resist CPMMV would enrich the introduction. Some of the examples I found are listed below, and pretty sure authors can enrich the introduction with more information.
https://doi.org/10.1111/pbr.12455
https://doi.org/10.2135/cropsci2012.01.0042
Also adding some discription (in terms of virus resistance) of the tested cultivars would be of good addition.

Experimental design

On the experimental level, the manuscript can be significantly improved with a few extra works to carry out:

1) Partial sequence may not give a sufficient idea about the distance to other isolates. I encourage authors to fully sequence the CPMMV isolate and generate a phylogenetic tree with other known isolates of CPMMV strains. For example, some ORFs of few soybean mosaic virus strains share 100% identity, such as those in G5H and G7H strain, but their strains’ pathogenicity is different in few cultivars. I encourage authors to fully sequence the isolate they have.

2) Test the susceptibility of those cultivars in lab conditions where temperature, light, humidity and other factors are fixed.

Validity of the findings

1) It will be of valuable addition if authors provide information about the climatic and geographical differences among the regions used in this study.

2) Compare the susceptibility of those cultivars depending on the region.

3) Please clarify the difference between symptomatic vs asymptomatic infection in terms of agronomic traits including crop yield.

Additional comments

I hope that authors address the experimental points in the revision so the manuscript can have better structure and results that are interesting for wider number of audience.

·

Basic reporting

The english language is clear and unambiguous. The manuscript structure is conform to Peer J. The summary is clear in succint. The number of tables and figures is appropriate and their quality adequate. The caption/legends contain sufficient information. The references are appropriate.

Experimental design

The introduction state the scientific problem clearly. The materials and methods used are adequate for the problem under investigation and sufficiently well documented. The statistical treatment is adequate.

Validity of the findings

The manuscript provide important new information on the effects of CPMMV to Brazilian soybean production and the abilty the Brazilian CPMMV to be transmitted by seed. The experimental procedures are competently conducted even if field trials were not repeated. Conclusions are well stated.

Additional comments

I consider the manuscript worthy for publication subject to the minor corrections and suggestions provided below. The title sound like a review and need to be improved.
Suggestions
Title : Effects of Cowpea mild mottle virus on soybean cultivars in Brazil
Lamas et al (Plant disease 101, 2017, Vol 10) shown that CPMMV was also detected in Sida sp (Malvaceae) and for the first time CPMMV was found in uncultivated plants belonging to several botanical families. Please include it in Lines 68-69 and complete your references.
Add one column to the table for the yield loss

---

## Round 0.2 · accepted · Accept

Thanks for bringing this interesting work to publication through PeerJ.